# A Generalist Neural Algorithmic Learner

**Borja Ibarz**[*]
DeepMind

**Vitaly Kurin**[†]
University of Oxford

**George Papamakarios**
DeepMind

**Kyriacos Nikiforou**
DeepMind

**Mehdi Bennani**
DeepMind

**Róbert Csordás**[†]
IDSIA, USI Lugano

**Andrew Dudzik**
DeepMind

**Matko Bošnjak**
DeepMind

**Alex Vitvitskyi**
DeepMind

**Yulia Rubanova**
DeepMind

**Andreea Deac**[†]
Mila, Université de Montréal

**Beatrice Bevilacqua**[†]
Purdue University

**Yaroslav Ganin**
DeepMind

**Charles Blundell**
DeepMind

**Petar Veličković**[*]
DeepMind

## Abstract

The cornerstone of neural algorithmic reasoning is the ability to solve algorithmic tasks, especially in a way that generalises out of distribution. While recent years have seen a surge in methodological improvements in this area, they mostly focused on building *specialist* models. Specialist models are capable of learning to neurally execute either only one algorithm or a collection of algorithms with identical control-flow backbone. Here, instead, we focus on constructing a *generalist* neural algorithmic learner—a single graph neural network processor capable of learning to execute a wide range of algorithms, such as sorting, searching, dynamic programming, path-finding and geometry. We leverage the CLRS benchmark to empirically show that, much like recent successes in the domain of perception, generalist algorithmic learners can be built by "incorporating" knowledge. That is, it is possible to effectively learn algorithms in a multi-task manner, so long as we can learn to execute them well in a single-task regime. Motivated by this, we present a series of improvements to the input representation, training regime and processor architecture over CLRS, improving average single-task performance by over $20\%$ from prior art. We then conduct a thorough ablation of multi-task learners leveraging these improvements. Our results demonstrate a generalist learner that effectively incorporates knowledge captured by specialist models.

## 1 Introduction

Machine learning systems based on deep neural networks have made tremendous strides in recent years, especially so for tasks dominated by *perception*. Prominent models in this space are usually required to generalise *in-distribution*, meaning that their training and validation sets are representative of the distribution expected of test inputs. In contrast, to truly master tasks dominated by *reasoning*, a model needs to provide sensible outputs even when generalising *out-of-distribution (OOD)*. Correspondingly, neural networks have seen lesser levels of success in this domain. Indeed, it has been suggested that stronger neural reasoning architectures may require careful application of methods such as algorithmic alignment [1], causality [2] and self-supervised learning [3]. Furthermore, these kinds of architectures are likely to be critical for robustly generating new knowledge based on existing observations, especially when that knowledge escapes the domain of training data.

---

[*]Corresponding authors. {bibarz,petarv}@deepmind.com

[†]Work performed while the author was at DeepMind.

B. Ibarz et al., A Generalist Neural Algorithmic Learner. *Proceedings of the First Learning on Graphs Conference (LoG 2022)*, PMLR 198, Virtual Event, December 9–12, 2022.

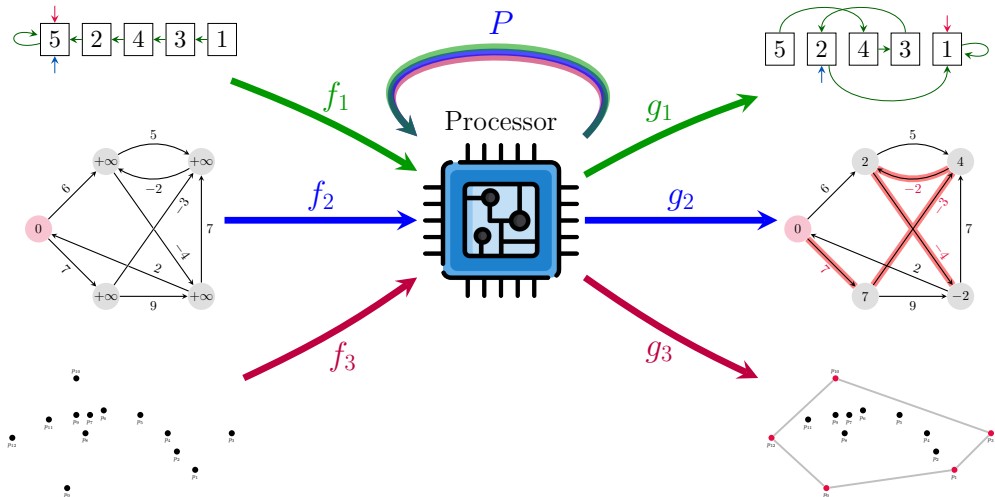

**Figure 1:** Our generalist neural algorithmic learner is a *single* processor GNN $P$, with a single set of weights, capable of solving several algorithmic tasks, $\tau$, in a shared latent space (each of which is attached to $P$ with simple encoders/decoders $f_\tau$ and $g_\tau$). Among others, our processor network is capable of sorting (**top**), shortest path-finding (**middle**), and convex hull finding (**bottom**).

Neural algorithmic reasoning [4] offers a robust route for obtaining such modelling advancements. Its focus is on evaluating existing (graph) neural network architectures on their ability to solve algorithmic tasks, typically by learning to execute classical algorithms [5]. This is an excellent target for probing reasoning capabilities, as classical algorithms can be seen as the essential "building blocks" for all of theoretical computer science, and fundamental tools in a software engineering career [6]. While this is a fairly self-contained pipeline, evidence of its applicability has already emerged: Graph Neural Networks (GNNs) pre-trained on algorithmic tasks have been successfully utilised in implicit planning [7] and self-supervised learning [8]. All of the prior advances in this area focused on building *specialist* models: either focusing on a single algorithm, or a collection of algorithms with an identical control flow backbone [9, 10].

In contrast, here we demonstrate a *generalist* neural algorithmic learner: a single GNN, with a single set of parameters, capable of learning to solve several classical algorithmic tasks simultaneously—to a level that matches relevant specialist models on average. This represents an important milestone, showing we can meaningfully incorporate reasoning capabilities even across tasks with completely disparate control flow, and in several tasks, we can exceed the OOD performance (performance on larger-size instances of the tasks) of the corresponding single-task specialist. Our generalist model is capable of performing various tasks, spanning sorting, searching, greedy algorithms, dynamic programming, graph algorithms, string algorithms and geometric algorithms (Figure 1). The experimentation we conduct is made possible by the CLRS-30 benchmark [5], a collection of thirty classical algorithmic tasks [6] spanning the above categories, along with a unified representational interface which made multi-task models easier to deploy.

Our results are powered by a single salient observation: any numerical difficulties which would make individual algorithms harder to learn (e.g. unstable gradients) are *amplified* when trying to learn a collection of such algorithms at once. Therefore, one of our main contributions is also to present a series of improvements to the training, optimisation, input representations, and GNN architectures which, taken together, improve the best-known average performance on the CLRS-30 benchmark by over 20% in absolute terms. We hope that our collection of improvements, with careful explanation for their applicability, will prove useful to GNN practitioners even beyond the realm of reasoning.

Following the overview of related work in Section 2, we describe, in Section 3, the improvements in the representation, training regime and architecture that lead to a single model with significantly better performance than previous published state-of-the-art (SOTA) on CLRS-30. We then show in Section 4, as our main contribution, that this model, trained simultaneously on all the CLRS-30 tasks, can match corresponding specialist models on average, demonstrating general algorithmic learning.

## 2 Related Work

The closest related work to ours is NeuralExecutor++, a multi-task algorithmic reasoning model by Xhonneux et al. [10, NE++]. NE++ focuses on a highly *specialised* setting where all the algorithms have an identical control flow backbone. For example, NE++ jointly learns to execute Prim's [11] and Dijkstra's [12] algorithms, which are the same up to a choice of key function and edge relaxation subroutine. Even in this specialist regime, the authors are able to make critical observations, such as empirically showing the specific forms of multi-task learning necessary for generalising OOD. We leverage these insights and extend them beyond the domain of closely related algorithms.

Also of note is the work on neural execution of graph algorithms by Veličković et al. [9]. This work provided early evidence of the potential for multi-task learning of classical algorithms. The authors simultaneously learn breadth-first search and the Bellman-Ford algorithm [13]—empirically demonstrating that joint learning is better than learning them either in isolation or with various forms of curriculum [14]. Once again, the algorithms have nearly-identical backbone; in fact, breadth-first search can be interpreted as the Bellman-Ford algorithm over a graph with constant edge weights.

Our work belongs to the *hard parameter sharing* class of models, pioneered by Caruana [15]. In hard parameter sharing, all tasks share the same model, with, potentially, some task-specific weights. This line of work has demonstrated that a single general model can learn a set of challenging tasks in combinatorial optimisation [16–18], computer control [19], and multi-modal multi-embodied learning [20, Gato]. Just like Gato provides a generalist agent for a wide variety of tasks (language modelling, playing Atari games, robotic control, image captioning), we provide a generalist agent for a diverse set of algorithmic domains, including sorting, searching, graphs, strings, and geometry.

Due to their ability to operate on graphs of arbitrary size, GNNs (including Transformers [21]) have been extensively explored for their in- and out-of-distribution generalisation properties in Reinforcement Learning (RL) [22–26]. In our setting, OOD generalisation implies generalisation to problems of larger size, e.g., longer input arrays to sort or larger graphs to find shortest paths in. In-distribution generalisation implies generalisation to new instances of problems of the same size. From this perspective, our problem setting is similar to procedurally-generated environments in RL [27–29].

The improvements we implemented for our single-task specialist reasoners are largely motivated by the theory of algorithmic alignment [30]. The key result of this theory is that neural networks will have provably smaller sample complexity if they are designed with components that "line up" with the target algorithm's operations. Following this prescription, we make several changes to the input data representations to make this alignment stronger [1], modify the GNN architecture to support higher-order reasoning [31] and suggest dedicated decoders for doubly-stochastic outputs [32].

## 3 Single-task experiments

Each algorithm in the CLRS benchmark [5] is specified by a number of *inputs*, *hints* and *outputs*. In a given sample, the inputs and outputs are fixed, while hints are time-series of intermediate states of the algorithm. Each sample for a particular task has a size, $n$, corresponding to the number of nodes in the GNN that will execute the algorithm.

A sample of every algorithm is represented as a graph, with each input, output and hint located in either the nodes, the edges, or the graph itself, and therefore has shape (excluding batch dimension, and, for hints, time dimension) $n \times f$, $n \times n \times f$, or $f$, respectively, $f$ being the dimensionality of the feature, which depends on its *type*. The CLRS benchmark defines five types of features: `scalar`, `categorical`, `mask`, `mask_one` and `pointer`, with their own encoding and decoding strategies and loss functions—e.g. a `scalar` type will be encoded and decoded directly by a single linear layer, and optimised using mean squared error. We defer to the CLRS benchmark paper [5] for further details.

### 3.1 Base Model

**Encoder.** We adopt the same *encode-process-decode* paradigm [33] presented with the CLRS benchmark [5]. At each time step, $t$, of a particular task $\tau$ (e.g. insertion sort), the task-based *encoder* $f_\tau$, consisting of a linear encoder for each input and hint, embeds inputs and the current hints as high-dimensional vectors. These embeddings of inputs and hints located in the nodes all have the same dimension and are *added* together; the same happens with hints and inputs located in edges, and in the graph. In our experiments we use the same dimension, $h = 128$, for node, edge and graph

embeddings. Thus, at the end of the encoding step for a time-step $t$ of the algorithm, we have a single set of embeddings $\left\{ \mathbf{x}_i^{(t)}, \mathbf{e}_{ij}^{(t)}, \mathbf{g}^{(t)} \right\}$, shapes $n \times h$, $n \times n \times h$, and $h$, in the nodes, edges and graph, respectively. Note that this is independent of the number and type of the inputs and hints of the particular algorithm, allowing us to *share* this latent space across all thirty algorithms in CLRS. Further, note that at each step, the input encoding is fed directly to these embeddings—this *recall* mechanism significantly improves the model's robustness over long trajectories [34].

**Processor.** The embeddings are fed into a *processor* $P$, a GNN that performs one step of computation. The processor transforms the input node, edge and graph embeddings into *processed* node embeddings, $\mathbf{h}_i^{(t)}$. Additionally, the processor uses the processed node embeddings from the previous step, $\mathbf{h}_i^{(t-1)}$, as inputs. Importantly, the same processor model can operate on graphs of *any* size. We leverage the message-passing neural network [35, MPNN], using the $\max$ aggregation and passing messages over a *fully-connected graph*, as our base model. The MPNN computes processed embeddings as follows:

$$\mathbf{z}^{(t)} = \mathbf{x}_i^{(t)} \| \mathbf{h}_i^{(t-1)} \qquad \mathbf{m}_i^{(t)} = \max_{1 \leq j \leq n} f_m \left( \mathbf{z}_i^{(t)}, \mathbf{z}_j^{(t)}, \mathbf{e}_{ij}^{(t)}, \mathbf{g}^{(t)} \right) \qquad \mathbf{h}_i^{(t)} = f_r \left( \mathbf{z}_i^{(t)}, \mathbf{m}_i^{(t)} \right) \quad (1)$$

starting from $\mathbf{h}^{(0)} = \mathbf{0}$. Here $\|$ denotes concatenation, $f_m : \mathbb{R}^{2h} \times \mathbb{R}^{2h} \times \mathbb{R}^h \times \mathbb{R}^h \to \mathbb{R}^h$ is the *message function* (for which we use a three-layer MLP with ReLU activations), and $f_r : \mathbb{R}^{2h} \times \mathbb{R}^h \to \mathbb{R}^h$ is the *readout function* (for which we use a linear layer with ReLU activation). The use of the $\max$ aggregator is well-motivated by prior work [5, 9], and we use the fully connected graph—letting the neighbours $j$ range over all nodes ($1 \leq j \leq n$)—in order to allow the model to overcome situations where the input graph structure may be suboptimal. Layer normalisation [36] is applied to $\mathbf{h}_i^{(t)}$ before using them further. Further details on the MPNN processor may be found in Veličković et al. [5].

**Decoder.** The processed embeddings are finally decoded with a task-based *decoder* $g_\tau$, to predict the hints for the next step, and the outputs at the final step. Akin to the encoder, the task-based decoder relies mainly on a linear decoder for each hint and output, along with a mechanism to compute pairwise node similarities when appropriate. Specifically, the `pointer` type decoder computes a score, $s_{ij}$, for each pair of nodes, and then chooses the pointer of node $i$ by taking either the $\operatorname{argmax}_j s_{ij}$ or $\operatorname{softmax}_j s_{ij}$ (depending on whether a hard or soft prediction is used).

**Loss.** The decoded hints and outputs are used to compute the loss during training, according to their type [5]. For each sample in a batch, the hint prediction losses are averaged across hints and time, and the output loss is averaged across outputs (most algorithms have a single output, though some have two outputs). The hint loss and output loss are added together. Besides, the hint predictions at each time step are fed back as inputs for the next step, except possibly at train time if *teacher forcing* is used (see Section 3.2.1).

We train the model on samples with sizes $n \leq 16$, and periodically evaluate them on in-distribution samples of size $n = 16$. Also, periodically, we evaluate the model with the best in-distribution evaluation score so far on OOD samples of size $n = 64$. In what follows, we will be reporting only these OOD evaluation scores. Full details of the model, training and evaluation hyperparameters can be found in Appendix A.

## 3.2 Model improvements

As previously discussed, single-task improvements, especially in terms of learning stability, will empirically transfer well to multi-task algorithmic learning. We now describe, in a gradual manner, all the changes made to the model, which have lead to an absolute improvement of over $20\%$ on average across all 30 tasks in CLRS.

### 3.2.1 Dataset and training

**Removing teacher forcing.** At evaluation time, the model has no access to the step-by-step hints in the dataset, and has to rely on its own hint predictions. However, during training, it is sometimes advisable to stabilise the trajectories with *teacher forcing* [37]—providing the ground-truth hint values instead of the network's own predictions. In the prior model [5], ground-truth hints were

provided during training with probability $0.5$, as, without teacher forcing, losses tended to grow unbounded along a trajectory when scalar hints were present, destabilising the training. In this work we incorporate several significant stabilising changes (described in future paragraphs), which allows us to remove teacher forcing altogether, aligning training with evaluation, and avoiding the network becoming overconfident in always expecting correct hint predictions. With teacher forcing, performance deteriorates significantly in sorting algorithms and Kruskal's algorithm. Naïve String Matcher, on the other hand, improves with teacher forcing (see Appendix A, Figs. 7-9).

**Augmenting the training data.** To prevent our model from over-fitting to the statistics of the fixed CLRS training dataset [5], we augmented the training data in three key ways, without breaking the intended size distribution shift. Firstly, we used the on-line samplers in CLRS to generate new training examples on the fly, rather than using a fixed dataset which is easier to overfit to. Secondly, we trained on examples of mixed sizes, $n \leq 16$, rather than only 16, which helps the model anticipate for a diverse range of sizes, rather than overfitting to the specifics of size $n = 16$. Lastly, for graph algorithms, we varied the connectivity probability $p$ of the input graphs (generated by the Erdős-Rényi model [38]); and for string matching algorithms, we varied the length of the pattern to be matched. These both serve to expose the model to different trajectory lengths; for example, in many graph algorithms, the amount of steps the algorithm should run for is related to the graph's diameter, and varying the connection probability in the graph generation allows for varying the expected diameter. These changes considerably increase training data variability, compared to the original dataset in Veličković et al. [5]. We provide a more detailed step-by-step overview of the data generation process in Appendix A.

**Soft hint propagation.** When predicted hints are fed back as inputs during training, gradients may or may not be allowed to flow through them. In previous work, only hints of the `scalar` type allowed gradients through, as all categoricals were post-processed from logits into the ground-truth format via $\mathrm{argmax}$ or thresholding before being fed back. Instead, in this work we use $\mathrm{softmax}$ for `categorical`, `mask_one` and `pointer` types, and the logistic sigmoid for `mask` types. Without these soft hints, performance in sorting algorithms degrades (similarly to the case of teacher forcing), as well as in Naïve String Matcher (Appendix A, Figs. 7-9).

**Static hint elimination.** Eleven algorithms in CLRS[3] specify a fixed ordering of the nodes, common to every sample, via a node pointer hint that does not ever change along the trajectories. Prediction of this hint is trivial (identity function), but poses a potential problem for OOD generalisation, since the model can overfit to the fixed training values. We therefore turned this fixed hint into an input for these 11 algorithms, eliminating the need for explicitly predicting it.

**Improving training stability with encoder initialisation and gradient clipping.** The `scalar` hints have *unbounded* values, in principle, and are optimised using mean-squared error, hence their gradients can quickly grow with increasing prediction error. Further, the predicted `scalar` hints then get *re-encoded* at every step, which can rapidly amplify errors throughout the trajectory, leading to exploding signals (and consequently gradients), even before any training takes place.

To rectify this issue, we use the Xavier initialisation [45], effectively reducing the initial weights for `scalar` hints whose input dimensionality is just 1. However, we reverted to using the default LeCun initialisation [46] elsewhere. This combination of initialisations proved important for the initial learning stability of our model over long trajectories. Relatedly, in preliminary experiments, we saw drastic improvements in learning stability, as well as significant increases in validation performance, with gradient clipping [47], which we subsequently employed in all experiments.

### 3.2.2 Encoders and decoders

**Randomised position scalar.** Across all algorithms in the dataset, there exists a *position* scalar input which uniquely indexes the nodes, with values linearly spaced between $0$ and $1$ along the node index. To avoid overfitting to these linearly spaced values during training, we replaced them with random values, uniformly sampled in $[0, 1]$, sorted to match the initial order implied by the linearly spaced values. The benefit of this change is notable in algorithms where it would be easy to overfit to

---

[3]Binary Search, Minimum, Max Subarray [39], Matrix Chain Order, LCS Length, Optimal BST [40], Activity Selector [41], Task Scheduling [42], Naïve String Matcher, Knuth-Morris-Pratt [43] and Jarvis' March [44].

these positions, such as string matching. Namely, the model could learn to base all of its computations on the assumption that it will always be finding a $m$-character pattern inside an $n$-character string, even though at test time, $m$ and $n$ will increase fourfold.

**Permutation decoders and the Sinkhorn operator.** Sorting algorithms (Insertion Sort, Bubble Sort, Heapsort [48] and Quicksort [49]) always output a permutation of the input nodes. In the CLRS benchmark, this permutation is encoded as a `pointer` where each node points to its predecessor in the sorted order (the first node points to itself); this is represented as a $n \times n$ matrix $\mathbf{P}$ where each row is a one-hot vector, such that element $(i, j)$ is 1 if node $i$ points to node $j$. As with all types of pointers, such permutation pointers can be predicted using a row-wise softmax on unconstrained decoder outputs (logits), trained with cross entropy (as in Veličković et al. [5]). However, this does not explicitly take advantage of the fact that the pointers encode a permutation, which the model has to learn instead. Our early experiments showed that the model was often failing to predict valid permutations OOD.

Accordingly, we enforce a permutation inductive bias in the output decoder of sorting algorithms, as follows. First, we modify the output representation by rewiring the first node to point to the last one, turning $\mathbf{P}$ into a *permutation matrix*, i.e., a matrix whose rows *and* columns are one-hot vectors. We also augment the representation with a one-hot vector of size $n$ that specifies the first node, so we do not lose this information; this vector is treated like a regular `mask_one` feature. Second, we predict the permutation matrix $\mathbf{P}$ from unconstrained decoder outputs $\mathbf{Y}$ by replacing the usual row-wise softmax with the *Sinkhorn operator $\mathcal{S}$* [32, 50–53]. $\mathcal{S}$ projects an arbitrary square matrix $\mathbf{Y}$ into a *doubly stochastic* matrix $\mathcal{S}(\mathbf{Y})$ (a non-negative matrix whose rows and columns sum to 1), by exponentiating and repeatedly normalizing rows and columns so they sum to 1. Specifically, $\mathcal{S}$ is defined by:

$$\mathcal{S}^0(\mathbf{Y}) = \exp(\mathbf{Y}) \qquad \mathcal{S}^l(\mathbf{Y}) = \mathcal{T}_c(\mathcal{T}_r(\mathcal{S}^{l-1}(\mathbf{Y}))) \qquad \mathcal{S}(\mathbf{Y}) = \lim_{l \to \infty} \mathcal{S}^l(\mathbf{Y}), \qquad (2)$$

where $\exp$ acts element-wise, and $\mathcal{T}_r$ and $\mathcal{T}_c$ denote row and column normalisation respectively. Although the Sinkhorn operator produces a doubly stochastic matrix rather than a permutation matrix, we can obtain a permutation matrix by introducing a temperature parameter, $\tau > 0$, and taking $\mathbf{P} = \lim_{\tau \to 0^+} \mathcal{S}(\mathbf{Y}/\tau)$; as long as there are no ties in the elements of $\mathbf{Y}$, $\mathbf{P}$ is guaranteed to be a permutation matrix [52, Theorem 1].

In practice, we compute the Sinkhorn operator using a fixed number of iterations $l_{\max}$. We use a smaller number of iterations $l_{\max} = 10$ for training, to limit vanishing and exploding gradients, and $l_{\max} = 60$ for evaluation. A fixed temperature $\tau = 0.1$ was experimentally found to give a good balance between speed of convergence and tie-breaking. We also encode the fact that no node points to itself, that is, that all diagonal elements of $\mathbf{P}$ should be 0, by setting the diagonal elements of $\mathbf{Y}$ to $-\infty$. To avoid ties, we follow Mena et al. [53], injecting Gumbel noise to the elements of $\mathbf{Y}$ prior to applying the Sinkhorn operator, during training only. Finally, we transform the predicted matrix $\mathbf{P}$, and `mask_one` pointing to the first element, into the original pointer representation used by CLRS.

### 3.2.3 Processor networks

**Gating mechanisms.** Many algorithms only require updating a few nodes at each time step, keeping the rest unchanged. However, the MPNN we use (Equation 1) is biased towards the *opposite*: it updates *all* hidden states in each step. Although it is theoretically possible for the network to keep the states unchanged, learning to do so is not easy. With this in mind, and motivated by its effectiveness in NDRs [54], we augment the network with an *update gate*, biased to be closed by default. We found that the gate stabilizes learning on many of the tasks, and increases the mean performance over all tasks on single-task training significantly. Surprisingly, however, we did not find gating to be advantageous in the multi-task case.

To add gating to the MPNN model we produce a per-node gating vector from the same inputs that process the embeddings in Equation 1:

$$\mathbf{g}_i^{(t)} = f_g\left(\mathbf{z}_i^{(t)}, \mathbf{m}_i^{(t)}\right) \qquad (3)$$

where $f_g : \mathbb{R}^{2h} \times \mathbb{R}^h \to \mathbb{R}^h$ is the *gating function*, for which we use a two-layer MLP, with ReLU activation for the hidden layer and logistic sigmoid activation for the output. Importantly, the final layer bias of $f_g$ is initialized to a value of $-3$, which biases the network for not updating its

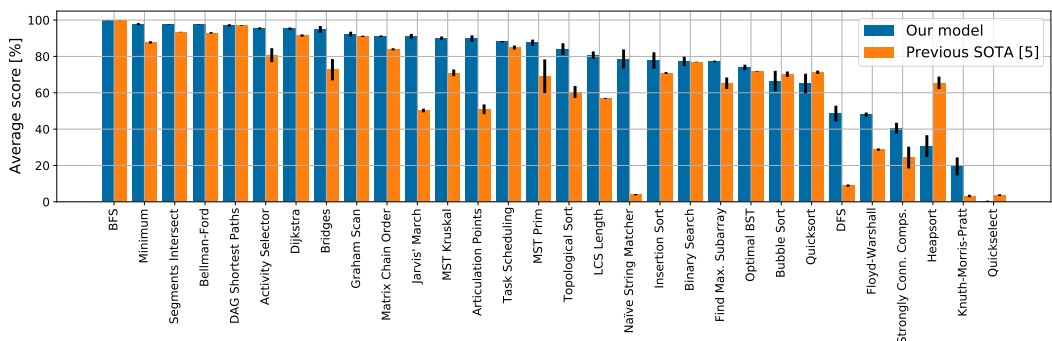

**Figure 2:** The OOD performance in single-task experiments before and after the improvements presented in this paper, sorted in descending order of current performance. Error bars represent standard error of the mean across seeds (3 seeds for previous SOTA experiments, 10 seeds for current). The previous SOTA values are the best of MPNN, PGN and Memnet models (see Table 2).

representations, unless necessary. The processed gated embeddings, $\widehat{\mathbf{h}}_i^{(t)}$, are computed as follows:

$$\widehat{\mathbf{h}}_i^{(t)} = \mathbf{g}_i^{(t)} \odot \mathbf{h}_i^{(t)} + (1 - \mathbf{g}_i^{(t)}) \odot \mathbf{h}_i^{(t-1)} \tag{4}$$

and are used instead of $\mathbf{h}_i^{(t)}$ in the subsequent steps, replacing $\mathbf{z}^{(t)}$ in Eq. 1 by $\mathbf{z}^{(t)} = \mathbf{x}_i^{(t)} \| \widehat{\mathbf{h}}_i^{(t-1)}$.

**Triplet reasoning.** Several algorithms within CLRS-30 explicitly require *edge-based reasoning*— where edges store values, and update them based on other edges' values. An example of this is the Floyd-Warshall algorithm [55], which computes all-pairs shortest paths in a weighted graph. The update rule for $d_{ij}$, its estimate for the best distance from node $i$ to $j$, is $d_{ij} = \min_k d_{ik} + d_{kj}$, which roughly says *"the best way to get from $i$ to $j$ is to find the optimal mid-point $k$, travel from $i$ to $k$, then from $k$ to $j$"*. Similar rules are pervasive across many CLRS-30 algorithms, especially in dynamic programming. Even though there are no *node* representations in the above update, all our processors are centered on passing messages between node representations $\mathbf{h}_i$.

To rectify this situation, we augment our processor to perform message passing towards edges. Referring again to the update for $d_{ij}$, we note that the edge representations are updated by choosing an intermediate node, then aggregating over all possible choices. Accordingly, and as previously observed by Dudzik and Veličković [31], we introduce *triplet reasoning*: first, computing representations over triplets of nodes, then reducing over one node to obtain edge latents:

$$\mathbf{t}_{ijk} = \psi_t(\mathbf{h}_i, \mathbf{h}_j, \mathbf{h}_k, \mathbf{e}_{ij}, \mathbf{e}_{ik}, \mathbf{e}_{kj}, \mathbf{g}) \qquad \mathbf{h}_{ij} = \phi_t(\max_k \mathbf{t}_{ijk}) \tag{5}$$

Here, $\psi_t$ is a *triplet message function*, mapping all relevant representations to a single vector for each triplet of nodes, and $\phi_t$ is an *edge readout function*, which transforms the aggregated triplets for each edge for later use. According to prior findings on the CLRS benchmark [5], we use the max aggregation to obtain edge representations. The computed $\mathbf{h}_{ij}$ vectors can then be used in any edge-based reasoning task, and empirically they are indeed significantly beneficial, even in tasks where we did not initially anticipate such benefits. One example is Kruskal's minimum spanning tree algorithm [56], where we presume that access to triplet reasoning allowed the model to more easily sort the edges by weight, as it selects how to augment the spanning forest at each step.

In order to keep the footprint of triplet embeddings as lightweight as possible, we compute only 8-dimensional features in $\psi_t$. $\phi_t$ then upscales the aggregated edge features back to 128 dimensions, to make them compatible with the rest of the architecture. Our initial experimentation demonstrated that the output dimensionality of $\psi_t$ did not significantly affect downstream performance. Note that computing triplet representations has been a useful approach in general GNN design [57]—however, it has predominantly been studied in the context of GNNs over *constant* input features. Our study is among the first to verify their utility over reasoning tasks with well-specified initial features.

### 3.3 Results

By incorporating the changes described in the previous sections we arrived at a single model type, with a single set of hyper-parameters, that was trained to reach new state-of-the-art performance

**Table 1:** Single-task OOD micro-$F_1$ score of previous SOTA Memnet, MPNN and PGN [5] and our best model Triplet-GMPNN with all our improvements, after 10,000 training steps.

| Alg. Type | Memnet [5] | MPNN [5] | PGN [5] | Triplet-GMPNN (ours) |
|---|---|---|---|---|
| Div. & C. | $13.05\% \pm 0.14$ | $20.30\% \pm 0.85$ | $65.23\% \pm 4.44$ | $\mathbf{76.36\% \pm 1.34}$ |
| DP | $67.94\% \pm 8.20$ | $65.10\% \pm 6.44$ | $70.58\% \pm 6.48$ | $\mathbf{81.99\% \pm 4.98}$ |
| Geometry | $45.14\% \pm 11.95$ | $73.11\% \pm 17.19$ | $61.19\% \pm 7.01$ | $\mathbf{94.09\% \pm 2.30}$ |
| Graphs | $24.12\% \pm 5.30$ | $62.79\% \pm 8.75$ | $60.25\% \pm 8.42$ | $\mathbf{81.41\% \pm 6.21}$ |
| Greedy | $53.42\% \pm 20.82$ | $82.39\% \pm 3.01$ | $75.84\% \pm 6.59$ | $\mathbf{91.21\% \pm 2.95}$ |
| Search | $34.35\% \pm 21.67$ | $41.20\% \pm 19.87$ | $56.11\% \pm 21.56$ | $\mathbf{58.61\% \pm 24.34}$ |
| Sorting | $\mathbf{71.53\% \pm 1.41}$ | $11.83\% \pm 2.78$ | $15.45\% \pm 8.46$ | $60.37\% \pm 12.16$ |
| Strings | $1.51\% \pm 0.46$ | $3.21\% \pm 0.94$ | $2.04\% \pm 0.20$ | $\mathbf{49.09\% \pm 23.49}$ |
| Overall avg. | $38.88\%$ | $44.99\%$ | $50.84\%$ | $\mathbf{74.14\%}$ |
| > 90% | 0/30 | 6/30 | 3/30 | **11/30** |
| > 80% | 3/30 | 9/30 | 7/30 | **17/30** |
| > 60% | 10/30 | 14/30 | 15/30 | **24/30** |

on CLRS-30 [5]. Tables 1 and 2 show the micro-$F_1$ scores of our model, which we refer to as Triplet-GMPNN (an MPNN with gating and triplet edge processing), over the original CLRS-30 test set (computed identically to Veličković et al. [5], but with 10 repetitions instead of 3). Our baselines include the Memnet [58], MPNN [35] and PGN [59] models, taken directly from Veličković et al. [5]. Figure 2 displays the comparison between the improved model and the best model from Veličković et al. [5]. Our improvements lead to an overall average performance that is more than 20% higher (in absolute terms) compared to the next best model (see Table 1), and to a significant performance improvement in all but one algorithm family, compared to every other model. Further, our stabilising changes (such as gradient clipping) have empirically reduced the scale of our model's gradient updates across the 30 tasks, preparing us better for the numerical issues of the multi-task regime. We finally also note that though we do not show it in Tables 1 & 2, applying the same improvements to the PGN processor, leads to an increase in overall performance from $50.84\%$ (Table 1) to $69.31\%$.

There are two notable examples of algorithm families with significant OOD performance improvement. The first are geometric algorithms (Segments Intersect, Graham Scan [60] and Jarvis' March), now solved at approximately $94\%$ OOD, compared to the previous best of about $73\%$; the second being string algorithms (Knuth-Morris-Pratt and Naïve String Matcher) for which our model now exceeds $49\%$ compared to the previous best of approximately $3\%$.

The significant overall performance boost is reflected in the increased number of algorithms we can now solve at over $60\%$, $80\%$ & $90\%$ OOD performance, compared to previous SOTA [5]. Specifically, we now exceed $60\%$ accuracy in 24 algorithms (15 algorithms previously), $80\%$ for 17 algorithms (9 previously) and $90\%$ for 11 algorithms (6 previously).

## 4 Multi-task experiments

In the multi-task setting, we train a single processor across all CLRS-30 tasks. We keep encoders and decoders separate for each task. To perform the update, one might accumulate gradients from all the tasks before stepping the optimizer, or step independently after each batch from each algorithm. Both approaches have been deemed to be effective in the multi-task learning literature [20, 24, 61], and we empirically found that, in our setting, stepping separately per task produced superior results. Following recent work [61], we did not explore specialised multi-task optimizers, but ensured the stability of the training with gradient clipping [47] and Xavier initialisation [45] of scalar hint encoders to ameliorate exploding outputs and NaN gradients, as already described. Batch size and learning rate are the same as in single-task experiments. We found that gating (Section 3.2.3) degraded multi-task performance, so it was not included in the multi-task model.

**Chunking.** To reduce the memory footprint of multi-task training we implemented a *chunked* training mode, where trajectories are split along the time axis for gradient computation and, when they are shorter than the chunk length, are concatenated with the following trajectory so as to avoid

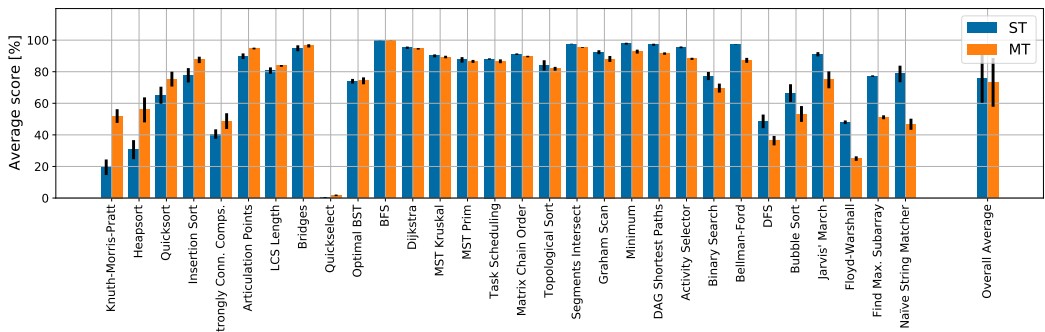

**Figure 3:** Per-algorithm comparison between our multi-task model and single-task Triplet-GMPNN from Table 2, ordered by biggest improvement for multi-task (left to right). Refer to Figure 5 for a comparison against the best single-task model per algorithm instead.

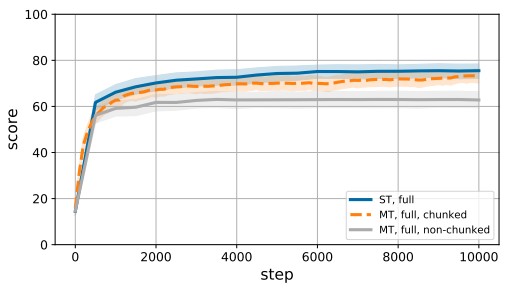

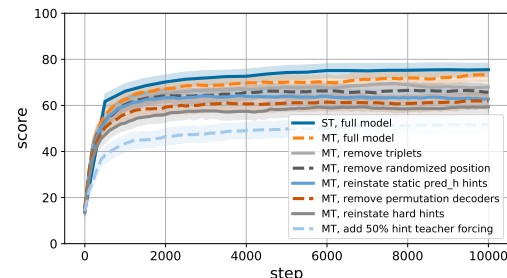

**(a)** Chunking significantly improves the multi-task model performance.

**(b)** Cumulative ablation demonstrates the positive effect of model improvements on the final OOD performance.

**Figure 4:** Multi-task model ablations showing average performance and 95% CI across 10 seeds. ST, single-task; MT, multi-task.

the need of padding. Thus, while a standard-training batch consists of full trajectories, padded to the length of the longest one, a chunked-training batch has a fixed time length (16 steps in our experiments) and consists of segments of trajectories. Immediately after the end of one trajectory the beginning of another one follows, so there is no padding. Losses are computed independently for each chunked batch, and gradients cannot flow between chunks. Since the output loss is computed only on the final sample of each trajectory, a chunk may give rise to no output loss, if it contains no end-of-trajectory segments. Chunking, therefore, changes the balance between hint and output losses depending on the length of trajectories. Surprisingly, multi-task performance averaged across all 30 tasks, after chunked training, is significantly better compared to full-trajectory training (Figure 4a). Only one algorithm, Bellman-Ford, has worse performance with chunked training (Figure 10). The strong effect of chunking on multi-algorithm performance indicates that the weighting of hint and output losses of the different tasks during optimization is important for successful multi-task learning.

**Results.** Figure 3 compares the performance of the single-task Triplet-GMPNN against the multi-task model. Additional comparisons against the best per-algorithms single-task model from Table 2 are also presented in Figure 5, along with an illustration of the number of tasks where the performance of multi-task model matches, or exceeds, that of single-task models. Finally, Figure 11 compares single-task and multi-task results against multi-task training on subsets of related algorithms.

To evaluate the effect of our model improvements independently, we also performed a thorough model ablation. Figure 4a shows the significant difference in performance between the vanilla and chunked training regimes; we chose the latter to perform the ablations on. Figure 4b shows the results of our *cumulative ablation*: we gradually removed our improvements one at a time, with each element in the legend being the same as the model preceding it with a single improvement removed. On average, all the presented improvements contribute to the higher performance, with the largest effect coming

from teacher forcing noise, i.e. feeding ground-truth hints at training time hurts generalisation, most likely because the correct hints are not available at test time, leading to data distribution shift.

# 5    Conclusion

We presented a *generalist* neural algorithmic learner: a single graph neural network, with a single set of weights, capable of solving a diverse collection classical algorithms, at a level comparable to (and at times exceeding) a relevant single-task expert. Achieving this objective was preceded by a range of improvements to the dataset, optimisation and architectures for neural algorithmic reasoning, which led to over 20% absolute improvements over the prior best known result. It is our hope that the results and empirical insights shared by this work will be of use to researchers and practitioners in the area, and help scale neural algorithmic learning to new domains and applications.

Contrary to implications of prior art [9, 10], our key takeaway is that it is indeed possible to learn diverse algorithms in a multi-task manner, but careful attention needs to be paid to the learning dynamics and stability of the (G)NN. Further, if modifications (to the GNN architecture, data pipeline, or loss functions) are made at the right level of generality, it is possible to improve algorithmic execution performance in large groups of algorithms at once. Lastly, the significant improvements obtained by chunking in the multi-task regime point that there are many interesting future avenues to explore on the utility of hint optimisation, and how it is counterbalanced with downstream output predictions—especially in the multi-task algorithmic learning regime.

## Author Contributions

The research idea of training a multi-task algorithmic reasoner was conceived and steered by Charles Blundell and Petar Veličković. The success of this idea rested on a flexible algorithmic benchmark with a unified representation space across diverse algorithms, which led to the development of CLRS-30, a project that was co-led by Petar. Borja Ibarz is the current technical lead of CLRS-30, and was responsible for devising, implementing and managing all developments related to training multi-task reasoners on it. Vitaly Kurin was in charge of the experimental pipeline for all multi-task experiments. The single-task model improvements, on which the multi-task reasoner rests, were driven by Borja, Petar, George Papamakarios, Kyriacos Nikiforou, Mehdi Bennani, Róbert Csordás, Andrew Dudzik, Matko Bošnjak and Alex Vitvitskyi. Specifically, George and Mehdi helped develop the Sinkhorn operator, Róbert implemented the gating mechanisms, Matko developed the gradient clipping experiments, Borja performed all the significant modification to the CLRS dataset detailed here, and Petar and Andrew implemented the triplet reasoning module. Kyriacos ran numerous targeted experiments that helped identify the key shortcomings of our models, and directly led to several important model modifications and updates. Additionally, Andrew implemented efficiency fixes to the CLRS benchmark which resulted in many-fold speedups to our multi-task training, enabling our runs to finish in time for the submission. Additional model variants have been implemented and evaluated by Yulia Rubanova, Andreea Deac and Beatrice Bevilacqua. In addition, Andreea advised the project on multi-task reasoning best practices, guided by the findings of her prior relevant work on NeuralExecutor++, which serves as a basis for our algorithmic learner. Charles, Petar and Yaroslav Ganin performed important advisory duties on this work. All authors contributed to co-writing the paper and responding to reviewer feedback.

## Acknowledgements

We would like to thank the developers of JAX [62] and Haiku [63]. We also thank Michela Paganini and Pete Battaglia for reviewing the paper prior to submission, and all our anonymous reviewers for their careful feedback, strengthening the paper significantly. Lastly, we thank all the past and present users of the CLRS benchmark, for their very insightful comments and suggestions. Especially, without the care given to CLRS development and maintenance by Adrià Puigdomènech Badia and David Budden, this research would likely have never been possible.

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

# A    Appendix

## A.1    Example data augmentation pipeline

We elaborate on the procedure for generating a particular sample in our augmented training dataset.

Let's assume that we want to learn to execute an algorithm $A$. One training example trajectory for $A$ is generated as follows:

1. Choose a problem size, $4 \leq n \leq 16$, at random. In the case of string algorithms (Naïve String Matcher and Knuth-Morris-Pratt), $n$ is fixed at 20, and size randomness will come from the choice of needle length (see point 5 below).

2. Choose a connection probability, $p \in [0, 1]$, at random.

3. Generate an input, represented as a graph with $n$ nodes, and with input node, edge and graph features sampled to match the algorithm's spec (see Veličković et al. [5] for details on specs).

4. If the task is a graph algorithm, for every pair of nodes $(u, v)$, decide whether to connect them with an edge by sampling $e_{uv} \sim \text{Bernoulli}(p)$. This is the Erdős-Rényi model, $\text{ER}(n, p)$ [38].

5. If the task is a string algorithm, choose a pattern length $1 \leq m \leq \lfloor \frac{n}{2} \rfloor$ at random. Then use the first $n - m$ nodes to represent the string to be searched (the *haystack*), and the remaining $m$ nodes as the pattern to be matched (the *needle*).

6. Execute $A$ on the resulting input, recording intermediate states, to obtain the training trajectory.

Steps 3 and 6 are shared with the original CLRS-30 benchmark generation pipeline [5]. All of the other steps are newly introduced by our work, with the purpose of avoiding overfitting to a rigid distribution. Specifically:

- We vary the *problem size*, $n$, to avoid overreliance on a particular size and/or particular positional embeddings. The original CLRS-30 dataset, in comparison, kept $n = 16$ ($n = 20$ for string matching algorithms) during training.

- We vary the *connection probability*, $p$, to avoid overreliance on a particular neighbourhood size. The original CLRS-30 dataset, in comparison, kept $p$ fixed during training. The exact value of $p$ varied depending on the algorithm; most used $p = 0.5$, but Articulation Points, Bridges and MST Kruskal used $p = 0.2$ to avoid very long trajectories. In our augmentations we have used $p \in [0, 0.5]$ for these 3 algorithms, as opposed to $p \in [0, 1]$ for the rest.

- We vary the *needle length*, $m$, to avoid overreliance on specific needle/haystack boundaries in string matching. The original CLRS-30 dataset, in comparison, kept $m = 4$ during training.

- Lastly, we generate the dataset in an *online manner*, providing the model with an infinite source of training data, to avoid overreliance on any particular fixed-size dataset. The original CLRS-30 dataset, in comparison, is a pre-generated dataset which is kept fixed.

## A.2    Additional experimental details

We use an embedding size $h = 128$ across all experiments. We train in batches of size 32 using an Adam optimizer [64] with learning rate 0.001, $\beta_1 = 0.9$, $\beta_2 = 0.999$, $\epsilon = 10^{-8}$, employing

gradient clipping by norm [47] with the clipping constant $c$ empirically set to $1.0$. In single-task experiments, we train for $10{,}000$ batches; in the multi-task experiments, we train for $10{,}000$ cycles of $30$ batches, one per algorithm. When using multiple training sizes (that is, everywhere except in no-data-augmentation ablations), each batch of each algorithm contains samples of the same size $n$, and the sizes for each algorithm cycle along the sequence $[4, 7, 11, 13, 16]$, except for string matching algorithms, where the training size is always $n = 20$ (variability is achieved by randomising the needle size, see below). When using *chunking* in multi-task experiments, batches have a fixed unroll length of $16$ steps; otherwise, each batch contains full-length samples. In chunked experiments it is important to keep separate values of the processor embeddings for each algorithm and training size, since unrolls are split in time and a new batch must start from the last-step embedding state of the same trajectories.

The trained model is evaluated periodically during training on samples of size $n = 16$ ($n = 20$ for string matching algorithms), and the best-performing model seen so far is evaluated on OOD samples. OOD refers to generalisation with respect to problem size; specifically, our OOD samples have size $n = 64$ ($n = 80$ for string matching). Only OOD performance is reported in this paper. The OOD data used for evaluation is sampled on-the-fly, drawn randomly at each evaluation, the number of samples being the same as in the CLRS benchmark [5]. The exception is Tables 1 and 2, where, for fair comparison, we used the fixed OOD samples from the CLRS dataset. We found no significant difference in evaluations with the fixed test data or on-the-fly samples.

When using randomised edge connection probabilities $p$ for data augmentation in graph algorithms (that is, in all experiments except the no-data-augmentation ablations), we sampled $p$ independently for each sample, uniformly from the set $\{0, 1, 0.2, 0.3, 0.4, 0.5, 0.6, 0.7, 0.8, 0.9\}$. However, for Articulation Points, Bridges and MST Kruskal we used a value of $p/2$, since otherwise, with dense graphs, the algorithms produce very long trajectories that would not fit in GPU memory. In Naïve String Matcher and Knuth-Morris-Pratt we randomised the length of the needle uniformly between $1$ and $8$.

As discussed in the main text, data augmentation via sizes, connection probabilities and needle lengths only applied to the training data. Evaluation always used the fixed parameters established in the CLRS benchmark.

### A.3 Additional experimental results

**Table 2:** Single-task OOD average micro-$F_1$ score of previous SOTA Memnet, MPNN and PGN [5] and our best model Triplet-GMPNN with all the improvements described in Section 3.

| Algorithm | Memnet [5] | MPNN [5] | PGN [5] | Triplet-GMPNN (ours) |
|---|---|---|---|---|
| Activity Selector | $24.10\% \pm 2.22$ | $80.66\% \pm 3.16$ | $66.80\% \pm 1.62$ | $\mathbf{95.18\% \pm 0.45}$ |
| Articulation Points | $1.50\% \pm 0.61$ | $50.91\% \pm 2.18$ | $49.53\% \pm 2.09$ | $\mathbf{88.32\% \pm 2.01}$ |
| Bellman-Ford | $40.04\% \pm 1.46$ | $92.01\% \pm 0.28$ | $92.99\% \pm 0.34$ | $\mathbf{97.39\% \pm 0.19}$ |
| BFS | $43.34\% \pm 0.04$ | $\mathbf{99.89\% \pm 0.05}$ | $99.63\% \pm 0.29$ | $99.73\% \pm 0.04$ |
| Binary Search | $14.37\% \pm 0.46$ | $36.83\% \pm 0.26$ | $76.95\% \pm 0.13$ | $\mathbf{77.58\% \pm 2.35}$ |
| Bridges | $30.26\% \pm 0.05$ | $72.69\% \pm 4.78$ | $51.42\% \pm 7.82$ | $\mathbf{93.99\% \pm 2.07}$ |
| Bubble Sort | $\mathbf{73.58\% \pm 0.78}$ | $5.27\% \pm 0.60$ | $6.01\% \pm 1.95$ | $67.68\% \pm 5.50$ |
| DAG Shortest Paths | $66.15\% \pm 1.92$ | $96.24\% \pm 0.56$ | $96.94\% \pm 0.16$ | $\mathbf{98.19\% \pm 0.30}$ |
| DFS | $13.36\% \pm 1.61$ | $6.54\% \pm 0.51$ | $8.71\% \pm 0.24$ | $\mathbf{47.79\% \pm 4.19}$ |
| Dijkstra | $22.48\% \pm 2.39$ | $91.50\% \pm 0.50$ | $83.45\% \pm 1.75$ | $\mathbf{96.05\% \pm 0.60}$ |
| Find Max. Subarray | $13.05\% \pm 0.08$ | $20.30\% \pm 0.49$ | $65.23\% \pm 2.56$ | $\mathbf{76.36\% \pm 0.43}$ |
| Floyd-Warshall | $14.17\% \pm 0.13$ | $26.74\% \pm 1.77$ | $28.76\% \pm 0.51$ | $\mathbf{48.52\% \pm 1.04}$ |
| Graham Scan | $40.62\% \pm 2.31$ | $91.04\% \pm 0.31$ | $56.87\% \pm 1.61$ | $\mathbf{93.62\% \pm 0.91}$ |
| Heapsort | $\mathbf{68.00\% \pm 1.57}$ | $10.94\% \pm 0.84$ | $5.27\% \pm 0.18$ | $31.04\% \pm 5.82$ |
| Insertion Sort | $71.42\% \pm 0.86$ | $19.81\% \pm 2.08$ | $44.37\% \pm 2.43$ | $\mathbf{78.14\% \pm 4.64}$ |
| Jarvis' March | $22.99\% \pm 3.87$ | $34.86\% \pm 12.39$ | $49.19\% \pm 1.07$ | $\mathbf{91.01\% \pm 1.30}$ |
| Knuth-Morris-Pratt | $1.81\% \pm 0.00$ | $2.49\% \pm 0.86$ | $2.00\% \pm 0.12$ | $\mathbf{19.51\% \pm 4.57}$ |
| LCS Length | $49.84\% \pm 4.34$ | $53.23\% \pm 0.36$ | $56.82\% \pm 0.21$ | $\mathbf{80.51\% \pm 1.84}$ |
| Matrix Chain Order | $81.96\% \pm 1.03$ | $79.84\% \pm 1.40$ | $83.91\% \pm 0.49$ | $\mathbf{91.68\% \pm 0.59}$ |
| Minimum | $86.93\% \pm 0.11$ | $85.34\% \pm 0.88$ | $87.71\% \pm 0.52$ | $\mathbf{97.78\% \pm 0.55}$ |
| MST-Kruskal | $28.84\% \pm 0.61$ | $70.97\% \pm 1.50$ | $66.96\% \pm 1.36$ | $\mathbf{89.80\% \pm 0.77}$ |
| MST-Prim | $10.29\% \pm 3.77$ | $69.08\% \pm 7.56$ | $63.33\% \pm 0.98$ | $\mathbf{86.39\% \pm 1.33}$ |
| Naïve String Matcher | $1.22\% \pm 0.48$ | $3.92\% \pm 0.30$ | $2.08\% \pm 0.20$ | $\mathbf{78.67\% \pm 4.99}$ |
| Optimal BST | $72.03\% \pm 1.21$ | $62.23\% \pm 0.44$ | $71.01\% \pm 1.82$ | $\mathbf{73.77\% \pm 1.48}$ |
| Quickselect | $1.74\% \pm 0.03$ | $1.43\% \pm 0.69$ | $\mathbf{3.66\% \pm 0.42}$ | $0.47\% \pm 0.25$ |
| Quicksort | $\mathbf{73.10\% \pm 0.67}$ | $11.30\% \pm 0.10$ | $6.17\% \pm 0.15$ | $64.64\% \pm 5.12$ |
| Segments Intersect | $71.81\% \pm 0.90$ | $93.44\% \pm 0.10$ | $77.51\% \pm 0.75$ | $\mathbf{97.64\% \pm 0.09}$ |
| SCC | $16.32\% \pm 4.78$ | $24.37\% \pm 4.88$ | $20.80\% \pm 0.64$ | $\mathbf{43.43\% \pm 3.15}$ |
| Task Scheduling | $82.74\% \pm 0.04$ | $84.11\% \pm 0.32$ | $84.89\% \pm 0.91$ | $\mathbf{87.25\% \pm 0.35}$ |
| Topological Sort | $2.73\% \pm 0.11$ | $52.60\% \pm 6.24$ | $60.45\% \pm 2.69$ | $\mathbf{87.27\% \pm 2.67}$ |
| Overall average | $38.03\%$ | $51.02\%$ | $52.31\%$ | $\mathbf{75.98\%}$ |

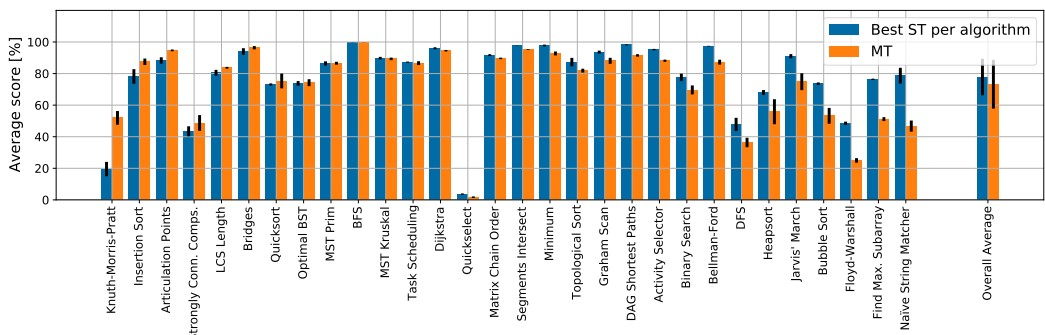

**(a)** Per-algorithm comparison between our multi-task model and the best per-algorithm model from Table 2, ordered by biggest improvement for multi-task (left to right).

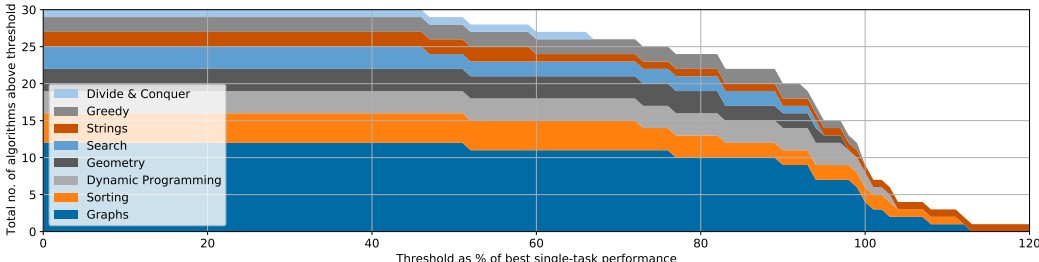

**(b)** Number of tasks where the performance of the multi-task model matched, or exceeded, a given percentage of the performance of the best single-task model (per algorithm) from Table 2, grouped by algorithm type. Note that, for some algorithms, the performance of the multi-task learner is higher than that of the best single-task learner.

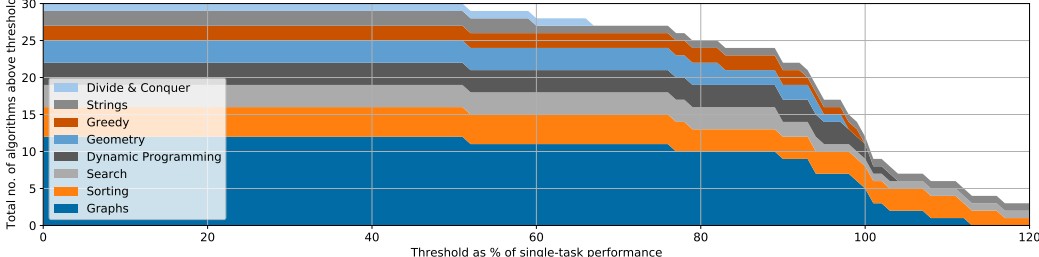

**(c)** Number of tasks where the performance of the multi-task model matched, or exceeded, a given percentage of the performance of single-task Triplet-GMPNN from Table 2, grouped by algorithm type.

**Figure 5:** Comparing our multi-task model to the best model per algorithm from Table 2 (5a & 5b). The comparison in 5c is between our multi-task model and our single-task Triplet-GMPNN.

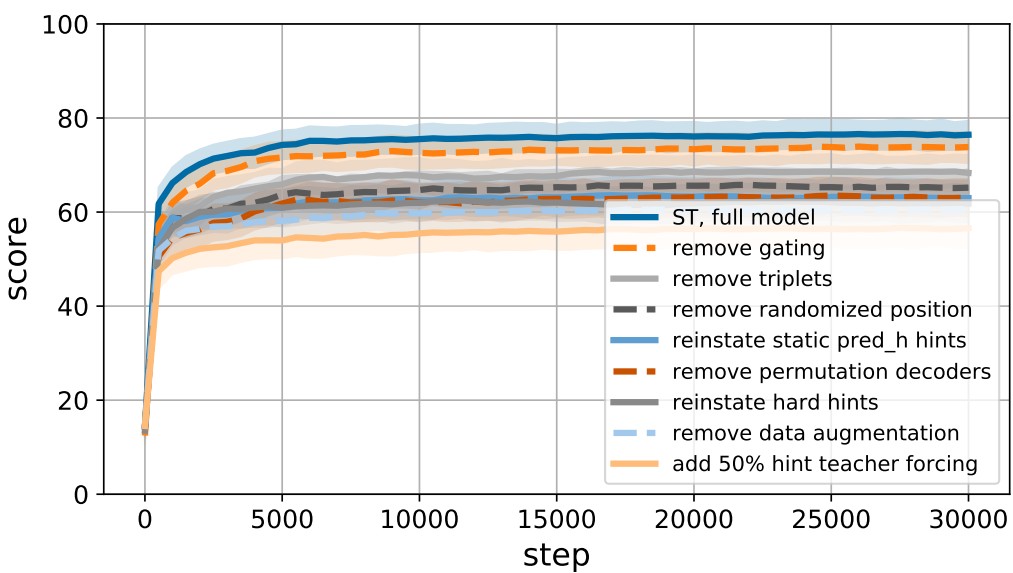

**Figure 6:** Single-task model cumulative ablations.

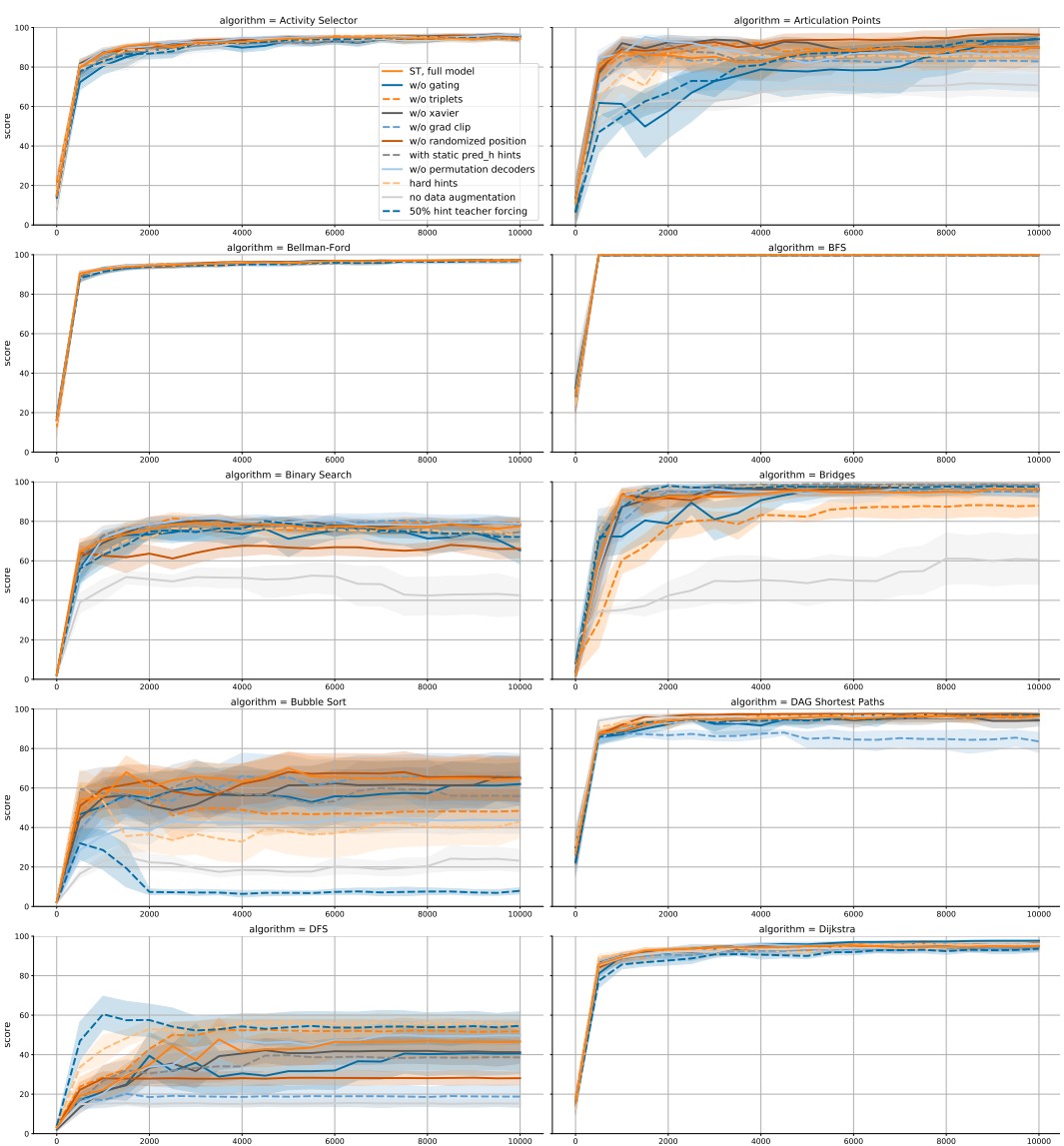

**Figure 7:** Non-cumulative single-task ablations faceted by algorithm. Part 1: Activity Selector, Articulation Points, Bellman-Ford, BFS, Binary Search, Bridges, Bubble Sort, DAG Shortest Paths, DFS and Dijkstra.

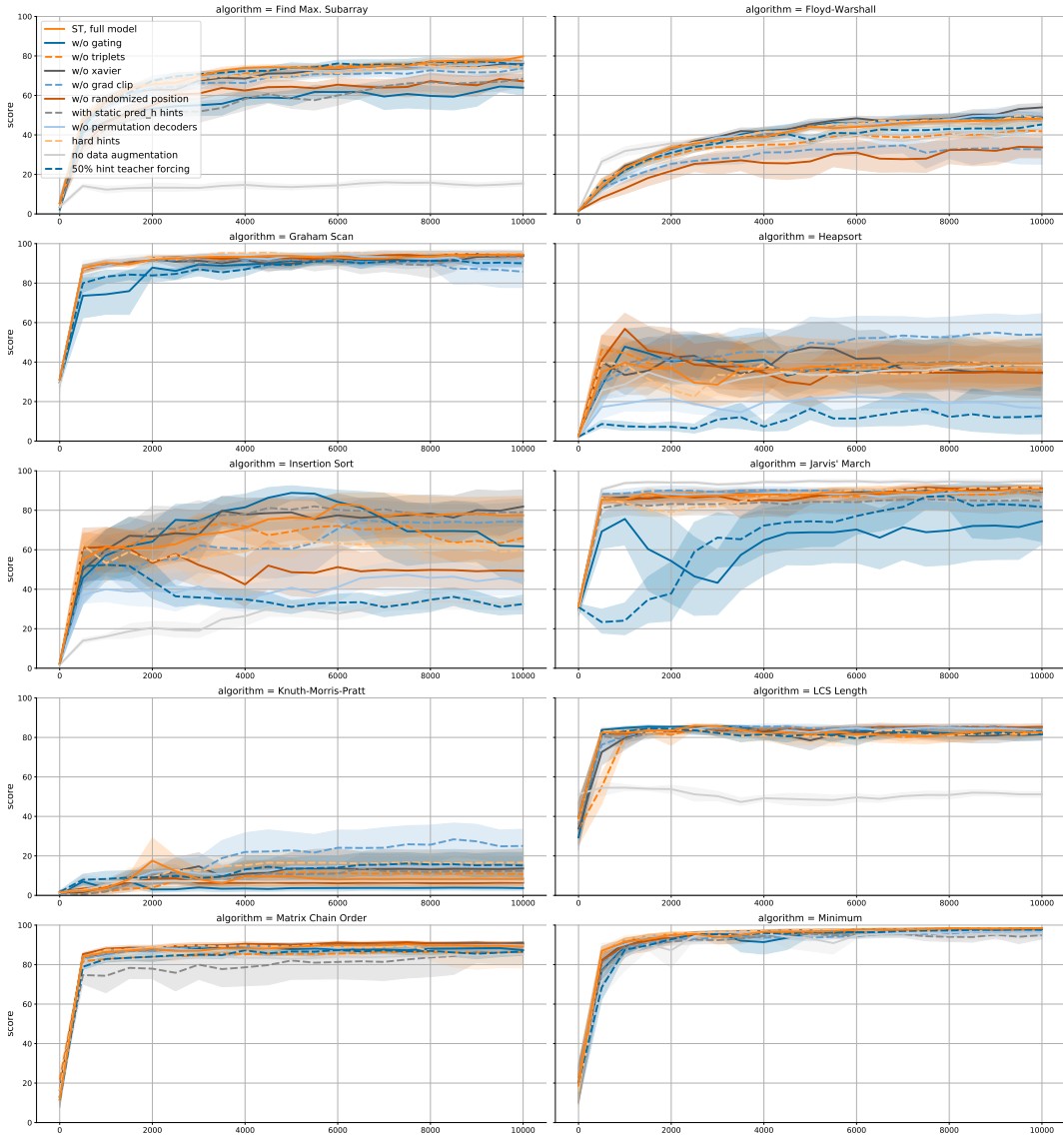

**Figure 8:** Non-cumulative single-task ablations faceted by algorithm. Part 2: Find Max. Subarray, Floyd-Warshall, Graham Scan, Heapsort, Insertion Sort, Jarvis' March, Knuth-Morris-Pratt, LCS Length, Matrix Chain Order and Minimum.

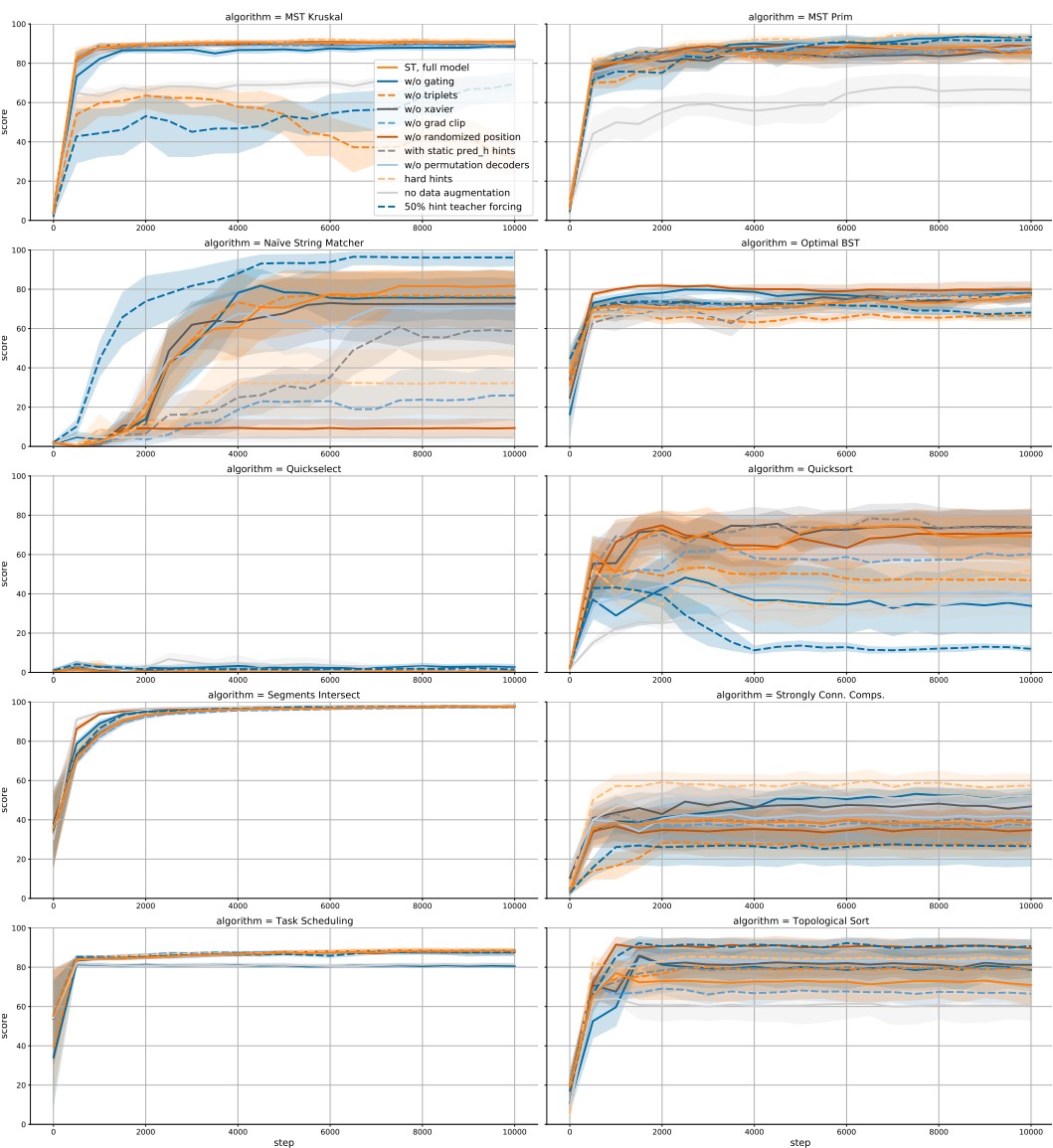

**Figure 9:** Non-cumulative single-task ablations faceted by algorithm. Part 3: MST-Kruskal, MST-Prim, Naïve String Matcher, Optimal BST, Quickselect, Quicksort, Segments Intersect, Strongly Connected Components, Task Scheduling and Topological Sort.

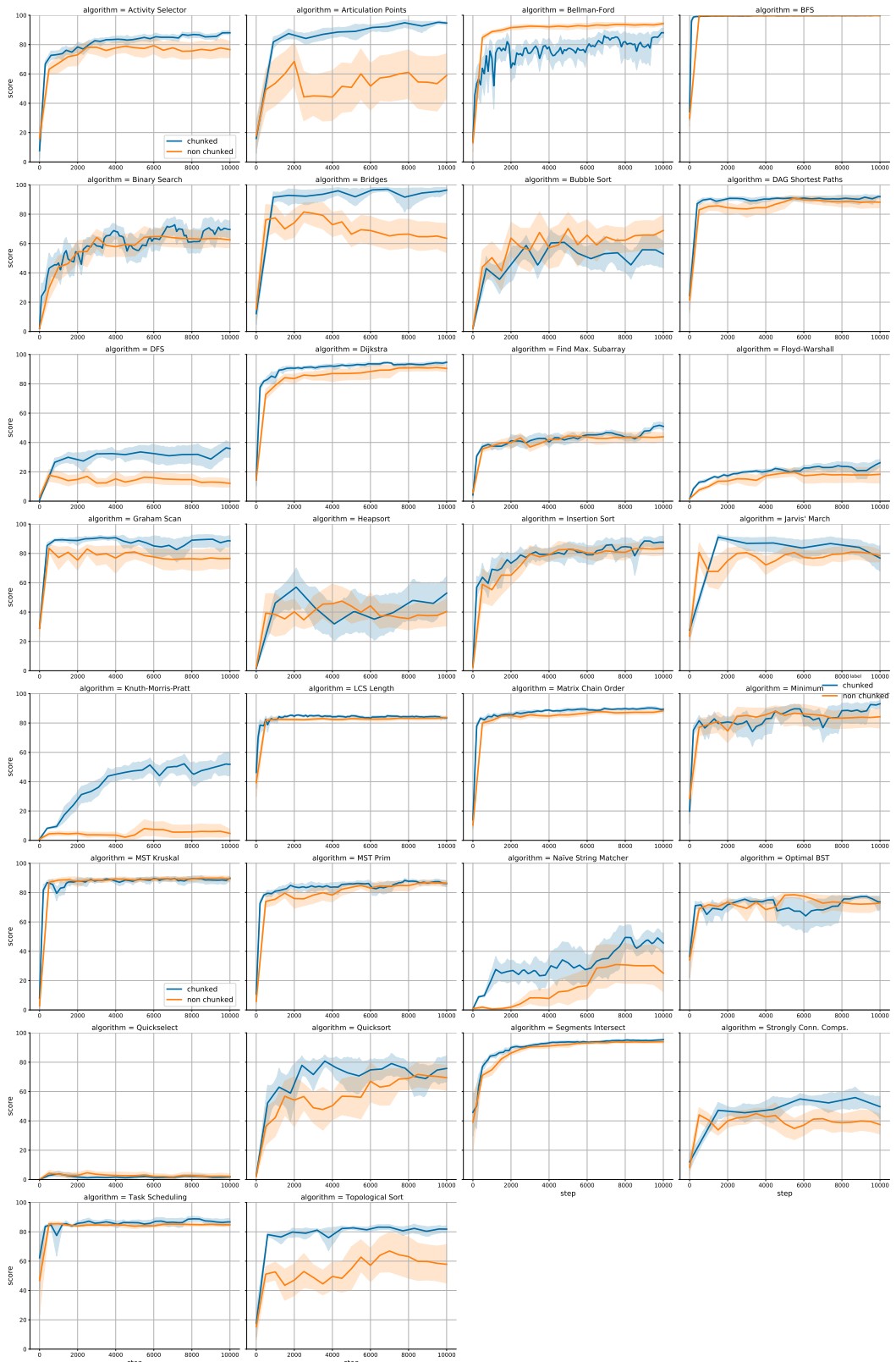

**Figure 10:** Per-algorithm comparison of chunked and non-chunked multitask models.

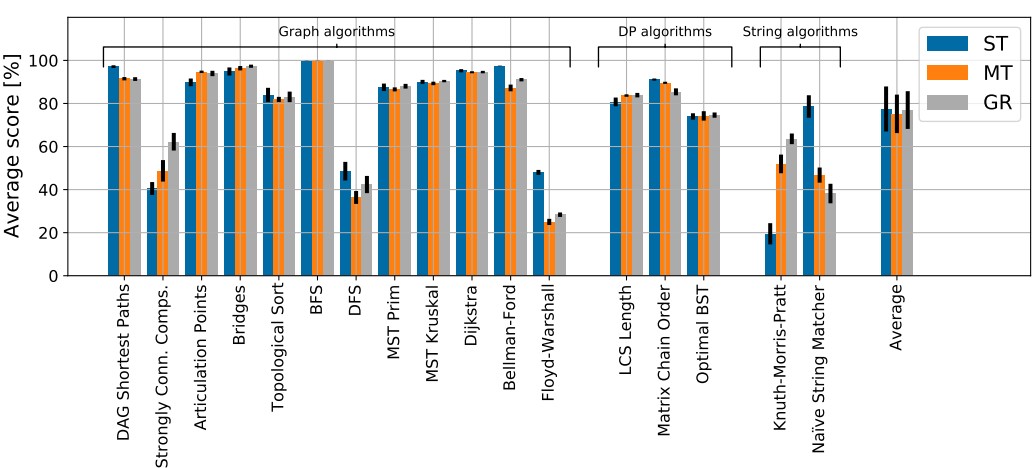

**Figure 11:** Per-algorithm comparison of multi-task and single-task training against training in groups of related algorithms (as per Veličković et al. [5]).

