# OpenReview forum: "A Generalist Neural Algorithmic Learner"
_logconference.io/LOG/2022/Conference — LoG 2022 Oral_

### Official Review · Reviewer_mLpP · 2022-10-21

**Overall Score:** 8
**Confidence:** 4

**Review:**

## Summary
 The paper proposes a single GNN model that advances the SOTA for neural algorithmic reasoning in the CLRS benchmark. Motivated by the notion of algorithmic alignment, the authors propose a series of modifications on a standard message passing pipeline that enables a model to be trained on multiple algorithmic tasks such as finding shortest paths and sorting. Apart from the significant improvements in SOTA for the CLRS benchmark, the paper demonstrates in certain cases that the performance of a single model trained on multiple tasks can exceed the performance of the same model trained on just one of those tasks.

## Strengths
- Significant improvements in out of distribution (OOD) performance across several tasks.
- The paper is clearly written, the results are presented in an intuitive way and the methodology is well explained.
- The paper provides thorough ablations for the architectural modifications that it proposes.
- Some of the enhancements described in the paper could be more broadly applicable in other settings beyond neural algorithmic reasoning.
- The fact that multi-task (MT) training is able to improve performance on certain tasks compared to single-task (ST) training is very interesting.
- The paper provides plausible explanations and arguments for the various techniques that are being employed.

## Weaknesses, questions, and suggestions
- I am genuinely impressed but also a bit skeptical by the improvements that come from the multi-task  learning. My skepticism comes from the architectural differences between the muli-task learner and the single-task one (e.g., gating). In the per-task ablations provided in the appendix, I cannot find an ablation on the single task model on sorting problems w.r.t. gating. So I'm not entirely certain whether the OOD benefits in some tasks are due to a potential regularization effect from the multi-task training or from minor architectural changes that impede the single-task learner in some of the tasks.
Could it be that some of the changes in performance going from ST to MT are just due to these architectural differences, or no? I am trying essentially to understand whether the 'generalist' approach is truly beneficial and to what extent.
Could one get the same MT performance or better by doing ST with just slightly varying some of the architectural components across tasks?
In any case, it's still impressive that one model could learn to consistently do well across many tasks and I understand that trying all possible combinations of components for the ST setting would be extremely time consuming.

- Since the architectural decisions are motivated by algorithmic alignment, let me build on the previous point and attempt to generalize the question at a higher level: could there be competing alignment interests? In other words, could it be that certain architectural decisions that will line up well with certain algorithms will *necessarily* be counter-productive for other algorithms? Since alignment appears to be the main principle championed by the paper, I am curious what its limitations are in this kind of multi-task setting.

- A concern that stems from the previous points is that the distribution of problems in the benchmark could be favoring certain kinds of architectural biases. If many problem categories can be solved very well by a certain execution paradigm, e.g., dynamic programming (DP),  one may end up having a 'generalist' with a blindspot for a certain problem category. It seems like sorting problems are particularly tough for the proposed model and the performance changes from ST to MT as well. The Memnet baseline seems to be doing well on that category but not doing well on the other problems. Is there a plausible explanation for why that is?
 On that note, performance in heapsort seems quite low. Can the authors speculate as to why this kind of degradation is pronounced only on one type of sorting algorithm?

- Does MT require a larger model than ST? I would be interested to see, for example, how ST and MT performance vary as the size of the model changes. Naively, I would expect that you need a larger model for the MT setting in order to facilitate learning all those different modalities of data at the same time. However, it could be interesting if the MT learner ends up being more parameter-efficient due to the diversity of the trainining.

- The standard deviation seems quite high in certain problems while very low on others. Could it be because OOD breaks after a certain size on some problems?
It would be interesting to know whether the model is just inconsistent on some problems regardless of instance size or whether there are 'phase transitions' in performance for certain sizes.
- What are potential limitations of the model enhancements that have been employed? As an example, triplet reasoning seems computationally expensive. Could the authors elaborate on the computational complexity of the model? Could there be bottlenecks if one were to scale those instances to much larger sizes?

- Overall, I'm not sure if any consistent conclusions for architecture design can be drawn from the paper. Some design choices appear to be beneficial but also circumstantial (e.g., chunking). Given the nature of the task this is understandable to some extent, but it could be beneficial if there was a 'key takeaways' paragraph that distilled all the empirical observations into some key takeaway points/observations (e.g., chunking seems to highlight the importance of the weighting of hint and output losses).




## Recommendation
Based on the overall results and the quality of presentation, I think this is a solid paper that should be accepted.  As I discussed above, the paper is not without its drawbacks, so I will start with a tentative score of 6 and I will update it after the authors' response.

## Post-rebuttal:
I have updated my score. See relevant comment.

---

### Official Review · Reviewer_HWv8 · 2022-10-21

**Overall Score:** 8
**Confidence:** 3

**Review:**

The paper presents a new method that allows a unique GNN processor (i.e. unique set of weights) to solve different algorithmic tasks on the CLRS benchmark. This is a novelty in the related literature where methods usually are able to provide a unique architecture that can solve all the tasks but requires retraining on each of the tasks.

STRONG POINTS:
- the research question is clear and the results support the main claim
- the paper is overall well written and easy to read / follow
- the paper contributes significantly in many technical and empirical aspects

IMPROVEMENT POINTS:
- the paper misses a broader motivation
- the OOD discussion/explanation/results are a bit confusing

I think the merits weight over the weaknesses, and I recommend a WEAK ACCEPT but I am willing to increase my score if the doubts I raised in the following comments are addressed by the authors.


Belowe some comments on the improvement points.


BROADER MOTIVATIONS: The paper clearly states the "narrow" motivations by highlighting the limitations of state-of-the-art models in the neural algorithmic reasoning task for the CLRS benchmark. However, as reading the paper, one is constantly faced with the implicit question "how far will this mod / this improvement go?"; "how generalizable are they to a broader context (e.g. to other similar tasks solved by GNNs, or to different algorithms)". Also within the CLRS, it would be interesting to know whether there are some of the choices/mods/improvements that are more foundational or whether some are more specific to the CLRS benchmark.


OOD: I think the writing could be polished w.r.t. the discussion about OOD generalization. The paper talks a lot about OOD generalization but the concurrent presence of a multi task setting made me think that OOD was related to generalization inter-task. Only in the related work the paper explains that OOD is referred to generalization to larger size problems. I think it is too delayed, as OOD generalization is used as an important motivation in the introduction, multiple times. Moreover, it is not clear how the OOD setting is tested in the experiments. The only mention to OOD is in the caption and then in the results. How is the OOD setting defined? Training examples? Test examples? Difference between them? Finally, it is not clear how much OOD generalization and the multi task training are interconnected.


Questions:
1) if you were asked to chose only two of the improvements you proposed on the base model, which you will point as more important?
2) (follow up of the previous question), how general are the proposed improvements within the CLRS benchmark?
3) can you give me some intuition about what is the impact/absence-of-impact of the augmentation of the data? I think this is a big change in a benchmark that would have required a bit more description / intuition / experimentation. Are the competitors equally augmented?
4) can you explain the experimental setting w.r.t. to OOD generalization? how is the OOD test built?
5) (follow up) Is the multi task training that improves OOD generalization? Or just the improvements over the base model already introduces improvements in OOD?




UPDATE: I thank the authors for the detailed explanations. I now have a better grasp on the OOD setting and I would like to raise my score to CLEAR ACCEPT.

---

### Official Review · Reviewer_T5tQ · 2022-10-21

**Overall Score:** 8
**Confidence:** 3

**Review:**

This paper reports on a specially designed GNN architecture extending on already established theory and ideas for solving algorithmic tasks with a high out-of-distribution generalization. The researchers already experimented with architectures that were task-specific and successful to a certain extent in similar tasks with comparable control flow, but the main goal is to reach a point where several different tasks can be solved by only one architecture. Related work and architectural decisions for the single and multi-task case are presented and in the end, the results and metrics help identify the benefits of this approach w.r.t. previous approaches. The authors show that in most cases their decisions were proven to be profitable.

The paper does contain significant content to justify a publication, although many of the ground-breaking ideas are in their previous research. Nevertheless, this publication brings an enormous amount of new decisions and heuristics to the table. Many of the questions that I will pose as a researcher is out of curiosity and for expanding my and other researchers' knowledge.

- Do you see any downsides to selecting a shared latent space across all 30 algorithms?
Will there be any considerations in the future to compare its representations with the latent spaces of the single-task algorithms?
- The augmentation of the training data, especially the last decision, is not straightforwardly clear. If there is space in the appendix please provide an example.
- Why do you think the model fails to predict valid permutations of OOD? Can you consider any way to validate your assumptions/interpretations and decide on further steps?
- Do you think that with the use of xAI methods you can explain phenomena such as f.e. gating not being advantageous in the multi-task case? What about explaining why some tasks like Heapsort or Bellman-Ford are better in the previous SOTA case? In the last sentence o the “Results” subsection, it would be quite interesting to see this interpretation being validated.

Overall this is very important research work from which we can learn a lot. I did not find any typos, the writing is excellent, and I would be even more excited about explanations of why some decisions were made beyond them being empirically validated.

---

### Official Review · Reviewer_Y4r6 · 2022-10-22

**Overall Score:** 8
**Confidence:** 4

**Review:**

The authors propose a set of improvements to the original neural algorithmic learner that results in a remarkable improvement on the CLRS benchmark. Each change is motivated by observations on the problem domain, the possible inductive biases present in original algorithms, and exploiting commonalities amongst algorithms.

Strengths of work
* Majority of the proposed changes are well-motivated and appear intuitive to the problem.
* The overall flow of the paper is great, and each technical aspect is clearly explained and stated.
* The research problem was well-stated and contributes to a significant upcoming area.

Weaknesses of work
* What is the point of graphs in the overall work? It was noted in line 135 that a complete graph was used, which essentially could render input structure useless. Could the authors comment on what if the original input graph structure was used? How then can we say that the processor network has indeed learnt to execute an algorithm if it instead is trained to ignore the structural biases of the problem?
* While the ablative studies were wide-ranged, what was the major difference in moving to a complete graph with MPNN? If that’s the case, what would a PGN with a complete graph look like?
* Since it appears that the impact of a complete graph is useful, I would like to suggest the authors provide some form of study on the impact of performance in terms of speed. While I understand that neural algorithmic reasoning is not meant to compete directly with very well-established algorithms, it would be nice to see the impact of their suggestions on the training and inference speeds as compared to previous attempts at the problem.
* While multi-task learning appeared to have improved the overall score, I am curious to see if we were to reduce the set of problems to those of a similar type (e.g. all sorting tasks), would the network do well? The general intuition would be that the network has learnt to do well on a task of algorithms. Would this at least be as good as completely training and learning on a large variety of tasks, of which some might have counterintuitive steps?

My recommendation is that the paper provides sufficient arguments for a clear acceptance. Each improvement was well-stated, the paper reads well, and the overall contribution to the neural algorithmic research community is significant. I would suggest the authors to provide some more details on the differences in using a complete graph vs. input graph. It could be that algorithms require some specific connections instead of the input graphs, and we should instead look on how to learn these structures.

---

### Meta-Review · Area_Chair_rATb · 2022-11-16

**Confidence:** 4
**Recommendation:** Accept for spotlight

**Meta Review:**

The reviewers have all agreed that the paper addresses an important research problem and demonstrates strong improvement over existing baselines. Moreover, some the of findings during the discussion period are interesting and may worth reporting for a broader community. As a result, the paper is clearly above the bar of publication at LOG conference.

---

### Decision · Program_Chairs · 2022-11-23

Accept (Oral)